# Research on the Real-Time Detection of Red Fruit Based on the You Only Look Once Algorithm

**Song Mei** [1,2], **Wenqin Ding** [1,*] **and Jinpeng Wang** [3]

1. Nanjing Institute of Agricultural Mechanization, Ministry of Agriculture and Rural Affairs, Nanjing 210014, China; meisong@caas.cn
2. College of Mechanical and Electrical Engineering, Nanjing University of Aeronautics and Astronautics, Nanjing 210016, China
3. Co-Innovation Center of Efficient Processing and Utilization of Forest Resources, Nanjing Forestry Univesity, Nanjing 210037, China; wangjinpeng@caas.cn
* Correspondence: dingwenqin@caas.cn

**Abstract:** The real-time and accurate recognition of fruits and vegetables is crucial for the intelligent control of fruit and vegetable robots. In this research, apple picking is selected. This study proposed a lightweight, coupon-product, neural-net terminal YOLO algorithm for apple image recognition. Compared with the YOLO series algorithm, the tiny algorithm shows a strong relationship with the calculation speed. In traditional red fruit detection, the recognition time is generally several seconds, which is unacceptable in the real-time system. In this research, a total of 2000 apple images from different environments are used as a dataset for training and testing. The YOLOv4-tiny model is detailed, instructed, and used for the identification. The indicators, such as F1Score (0.92) and mAP (95.5% average), are analyzed by calculating the loss rate, accuracy rate (96.21%), and recall rate (95.47%). Finally, the algorithm shows good accuracy and high speed (no more than 5 ms) in online real-time detection.

**Keywords:** deep learning; YOLO algorithm; dataset; real-time detection

## 1. Introduction

The ideas of deep learning [1,2] have played a key role in the rapid development of the field of machine vision. In recent years, object detection has developed to an unprecedented height, which is an important part of picture information processing and machine vision disciplines, and the core part of surveillance information systems. With the development of deep-learning theory and technology, the application scope of object detection is also greatly expanding. Some advanced algorithms and models can meet various complex needs, such as automatic driving, real-time tracking, intelligent fruit and vegetable harvesting, etc. Therefore, improving the accuracy and detection speed of the object detection algorithm can not only provide more accurate object category and position information for other downstream tasks, but also promote the application of these downstream tasks based on target detection, among which representative downstream tasks and their challenges to the target detection algorithm include the following:

(1) Pedestrian detection, especially small-scale pedestrian detection, is one of the challenges faced by object detection algorithms [3];

(2) Face detection, occlusion, and multi-scale object detection are also difficult challenges [4];

(3) Text detection and object detection for distorted, blurry, and low-quality images is one of the difficult problems that need to be solved [5];

(4) Fruit and vegetable testing [6,7]. Recently, many researchers have begun to explore new and more efficient techniques for detecting and identifying fruits and vegetables [8]. However, because traditional technologies cannot quickly and accurately

collect and analyze massive images, their robustness and accuracy still have certain shortcomings [9]. With the development of science and technology, new fruit and vegetable detection technology has been widely used in various scenarios. They can more accurately capture the appearance, structure, and properties of fruit, and can quickly and accurately identify the growth cycle of each fruit and the quality of the product. The application of these technologies has greatly improved the efficiency and accuracy of fruit and vegetable harvesting, bringing greater efficiency and faster harvest to agricultural production.

### 1.1. Review of Related Technologies for Red Fruit Detection

Red fruit detection is an important target detection technology for fruits and vegetables that can help us to better pick fresh fruits and vegetables, classify them, assess their ripe maturity, and detect pests and diseases. In recent years, research in this field has also been in full swing. In 2017, BARGOTI [10] developed a target detector based on Region-based Convolutional Neural Networks (R-CNNs) with Faster R-CNN that can accurately identify three different types of fruits and vegetables. In 2018, Peng Hongxing [11] used the deep convolutional network architecture of the Single-Shot MultiBox Detector (SSD) to identify a variety of fruits and vegetables. Li [12] used pretreatment technology to train Caffe Net and improved the recognition accuracy of strawberries to 95%. In 2019, Bi Song et al. [13] designed a method for citrus target recognition in the natural environment based on deep learning. Zeng Pingping [14] proposed a convolutional neural network model for identifying four fruits and vegetables: apples, pears, oranges, and peaches, which borrowed from the network architecture of Le Net. Cheng Hongfang et al. [15] proposed an improved LeNet apple detection model, which achieved a recognition rate of 93.79% for apple fruits in the natural environment. Huang Haojie [16] improved SSDs and realized the detection and classification of apples, oranges, and bananas. In 2020, Zhang Enyu [17] and his team developed a new detection model using SSD technology, which can effectively distinguish green apples in nature and has good accuracy. Gao F [18] and his team proposed a new apple fruit detection technology that uses Faster R-CNN technology to accurately divide the fruit according to the complex environment in which the apple fruit tree is located in its occlusion state to achieve more accurate fruit identification purposes.

### 1.2. Overview of Object Detection Based on Neural Networks

CNNs are an important means of image processing, which not only do not have to rely on complex models, but also have excellent generalization. CNNs [19,20] involve the layered processing of observed image information from the low layer to the high level. Then, the feature information of the image is processed in layers, and each layer processes specific information. After repeating this iteration many times, the low-level features are combined to form high-level features to gain a deep understanding of the observed objects. By adopting the weight allocation mechanism, the calculation of the model can be effectively simplified, and the computing resources can be effectively saved. When processing multiple forms of image information, convolutional neural networks can directly input data into the neural network to process images more effectively and better meet the needs of practical applications. In the process of processing image information, the convolutional neural network performs convolution calculations through the convolution kernel. The function of the convolution kernel is to find the characteristics of the picture. The convolution kernel is calculated from left to right, from top to bottom, and the extracted different feature pictures are obtained. Different picture features can be extracted by using different convolution kernels, and the values in each convolution kernel are automatically learned by the algorithm without manual setting [21].

Object detection algorithms can be roughly divided into two categories: two-stage algorithms and single-stage algorithms. The signature two-stage object detection algorithms are RCNN, the Spatial Pyramid Pooling Network (SPP-Net) [22], Faster R-CNN [23], Region-based Fully Convolutional Networks (R-FCNs), Mask R-CNN, Cascade RCNN, and Trident

Net. Representative single-stage object detection algorithms include SSD, You Only Look Once (YOLO), YOLO9000, Retina Net, You Only Look Once version 3 (YOLOv3), Efficient Det, and YOLOv4 [24].

In November 2013, Ross Girshick's R-CNN achieved great success in the field of object detection applications. However, to identify complex targets more accurately, the input image of the R-CNN must be processed according to a preset size, so this processing may lose the original data, which in turn affects the final inspection results.

In June 2014, He proposed a new SPP Net that reduces data loss due to image scaling by using SPP layers. However, SPP Net is similar to R-CNN in that it is inefficient to detect and must also face a large number of feature processing, which puts extremely high requirements on device performance.

In April 2015, Ross Girshick and other researchers proposed that Fast R-CNN can effectively reduce the complexity of object detection. So, it integrates multiple classification and regression steps into a single module, thereby greatly reducing the complexity of detection and greatly speeding up the detection process of the algorithm.

In June 2015, Redmon proposed YOLO, a single-stage object detection algorithm. It uses only one backbone process during the inspection process, which can significantly reduce computational complexity and memory usage and make it run faster. The invention of YOLO marks an important milestone. Although it is not as accurate as the two-stage detection algorithm, it opens a whole new path for academia to achieve a faster and more real-time detection method by simplifying the process. In December of that year, Liu and other scholars proposed the SSD algorithm, which combines the advantages of YOLO and explores its shortcomings. To achieve a balance, scholars introduced the anchor frame mechanism of the two-stage algorithm into the single-stage algorithm to improve the accuracy.

First proposed by He and other researchers in March 2017, the Mask R-CNN algorithm replaces the "ROI Pooling" layer of the Faster R-CNN algorithm with the "ROI Align" layer, and adds a new path, the "object mask". It can more accurately describe and identify specific objects, which is different from traditional object recognition methods. Due to the existence of the loss function, this leads to the loss calculation of the mask, which makes the calculation amount larger again, resulting in the difficulty of improving the efficiency of detection.

In August 2017, the Retina Net single-stage object detection method was developed by the Facebook AI Research team, and can effectively solve the time-consuming problem of two-stage modes. It can effectively avoid the decline in the accuracy of single-stage object detection due to the difference of multiple tasks.

In 2018, Redmon, the founder of YOLO, and his team pioneered the introduction of the YOLOv3 algorithm. It is based on the concept of FPN, combining three feature maps of different sizes to better achieve multi-scale object detection.

The excellent performance of the FPN algorithm has attracted many researchers, who draw on this algorithm and constantly explore and improve its application. In 2019, Tan et al. launched the Efficient Det algorithm, an improved version of FPN, namely the Bidirectional Cross-Scale Connection sand Weighted Feature Fusion Pyramid Networks. It greatly expanded the application scope of the BiFPN algorithm and brought new opportunities and challenges to the development of object detection algorithms. BiFPN enabled the effective detection of targets by fusing the features of both sides between the P3 layer and the P7 layer. Although the Efficient Det has many notable features and excellent accuracy, it is still necessary to improve the efficiency of its inspection.

With the widespread adoption of YOLOv3 technology, it has become one of the most common deep-learning object detection methods in use today. In 2020, Boch Kovskiy et al. launched YOLOv4, which replaces YOLOv3's Darknet53 network structure. It uses CSP Darknet53 to continue the essence of the YOLO algorithm, thereby promoting the progress of the YOLOv4 algorithm. The YOLOv4 method is a method that guarantees both speed and accuracy, and its unique features are its SPP module, PANet, and its many computing

tools. It can effectively combine different perception elements to better meet different needs. Therefore, it was the best object detection algorithm at that time, which guaranteed both fast and accurate results.

In recent years, deep learning techniques have shown promising results in the field of disease classification using image data. Five widely used deep learning models, namely AlexNet, VGG16, InceptionV3, MobileNetV3, and Efficient Net, were trained and evaluated using a dataset of sunflower disease images by Yonis et al. [25]. A well-known deep learning model, MobileNetV2, was used by Gulzar [26] as the base model but was modified by adding five different layers to improve the accuracy and reduce the error rate during the classification process. The proposed model is trained on a dataset containing 40 varieties of fruits and is validated to have the highest accuracy in recognizing different types of fruits. Poonam Dhiman [27] et al. described an SLR focused on disease identification and classification in citrus fruits using machine learning, deep learning, and statistical techniques and presented different conceptualized theories related to all the essential components of the recognition and classification of citrus fruit diseases. This SLR has addressed nearly all the state-of-the-art frameworks applicable to the detection of diseases in citrus fruits and also addressed stepwise measures to build a necessary automatic framework to protect fruits from apparent disease by answering nine research questions. Normaisharah Mamat et al. [28]. proposed an automatic image annotation advancement approach that employs repetitive annotation tasks to automatically annotate an object. The YOLOv5 model, a deep learning approach, is chosen for automatically annotating images. The design of this method is proven to be fast at annotating a new image, successfully achieves high accuracy, and can greatly reduce the amount of time required to classify fruit, while also addressing the difficulty caused by a massive number of unlabeled images.

Although current object detection technologies have made great progress and have reached a certain level, they are limited by computing power, environmental noise, daytime environments, different object sizes, and a variety of other environmental factors, which make them unable to achieve the desired results.

This study takes apples as an example of red fruits and vegetables as the research object to assist in the effective rapid detection of apples and the observation of the growth status of fruits and vegetables in real orchards. Considering the real scene and the more complex environment, we designed a lightweight neural network model to achieve the rapid detection of apples. The main research content of this study is as follows: in Section 2, this paper describes the overall design of the red fruit target detection scheme based on deep learning. In Section 3, YOLOv4-tiny is introduced. Section 4 shows the details of the YOLOv4-tiny implements. Section 5 discusses the results and Section 6 concludes the research.

## 2. Overall Design of Red Fruit Target Detection Scheme Based on Deep Learning

### 2.1. Algorithm Flow Based on Convolutional Neural Network

The training of convolutional neural networks includes two parts: The first stage is the forward propagation stage, in which the data flows from the input, layer by layer, from low level to high level. The second stage is the backpropagation stage, in which the error is transmitted from a high level to a low level when the calculated result does not match the expectation of the input. The training process is shown in Figure 1.

Through nonlinear transformation, the neural network can transform the original linear information into more complex nonlinear information, so that complex features can be learned and the expression ability of the model can be improved. By introducing the activation function, complex and changeable information can be effectively fused into the neural network model to strengthen the expression ability of the network. A few commonly used activation functions are as follows:

(1) Sigmoid activation function:

$$sig\,mod(x) = \frac{1}{1 + e^{-x}} \tag{1}$$

(2) Tanh activation function:

$$\tanh(x) = \frac{1 - e^{-2x}}{1 + e^{-2x}} \tag{2}$$

(3) Leaky ReLU activation function:

$$LeakyReLU(x) = \begin{cases} x & ,x > 0 \\ \alpha x & ,x \leq 0 \end{cases} \tag{3}$$

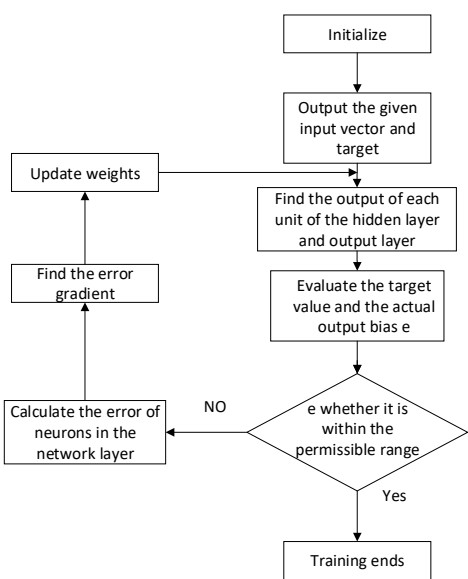

**Figure 1.** Neural network training flow chart.

### 2.2. Red Fruit Target Detection Scheme

As learned above, deep learning-based object detection algorithms include primary and secondary detection algorithms. Although some studies use two-stage detection algorithms for object detection in fruits and vegetables, such as Faster R-CNN and Mask R-CNN, the two-stage detection algorithm needs to invest more computing resources to select candidate regions. This also affects its detection efficiency in actual industrial environments, which affects the efficiency of their application to picking robots. In contrast, the one-stage detection algorithm can obtain accurate detection results faster and more in line with the needs of industrial scenarios.

After comparison, although the two-stage detection algorithm has higher accuracy than the first-stage detection algorithm, it has an extremely slow detection speed, a too-long training time, and is inefficient. Combined with this study, it is committed to red fruit target detection in industrial scenarios, and finally, we decided to use a one-stage detection algorithm to build a real-time detection function.

The one-stage detection algorithm [29,30] requires only one process to achieve detection. Feature extraction, target classification, and location regression are carried out in the whole convolutional network, and the target position and class can be obtained through one reverse calculation. Although the recognition accuracy is slightly weaker than that of the two-stage target detection algorithm, the speed is greatly improved. Typical one-stage detection algorithms include YOLO series algorithms, SSD, RetinaNet, etc.

The YOLO target detection process is to first resize the input image to $416 \times 416 \times 3$, then send it to the CNN network, and finally process the target detected by the network

prediction result. Compared with the R-CNN series of algorithms, it is a unified framework that has a faster speed, and its training process is end-to-end. YOLO is a widely used real-time object detection algorithm. Specifically, YOLO divides the whole graph into S × S lattices; each grid is responsible for detecting the targets falling into it and predicts the bounding box, confidence level, and probability vector of all the targets contained in all the lattices at once.

## 3. Algorithm Flow Based on YOLOv4-Tiny Architecture

Compared with SSD, other series of YOLO, and other series of one-stage detection algorithms, YOLOv4-tiny is a lightweight version of YOLOV4, with fewer layers and a faster detection speed. It can be used in portable devices, requires less GPU resources for training, and has a low memory load, which can be widely used in industrial sites. Therefore, this study adopts the YOLOv4-tiny architecture, and its core architecture can be seen in Figure 2, which is divided into two parts: CSPdark-net53_tiny and FPN. CSPdarknet53_tiny classifies and predicts image candidate boxes and FPN performs feature extraction of different sizes to see whether the network contains the target feature.

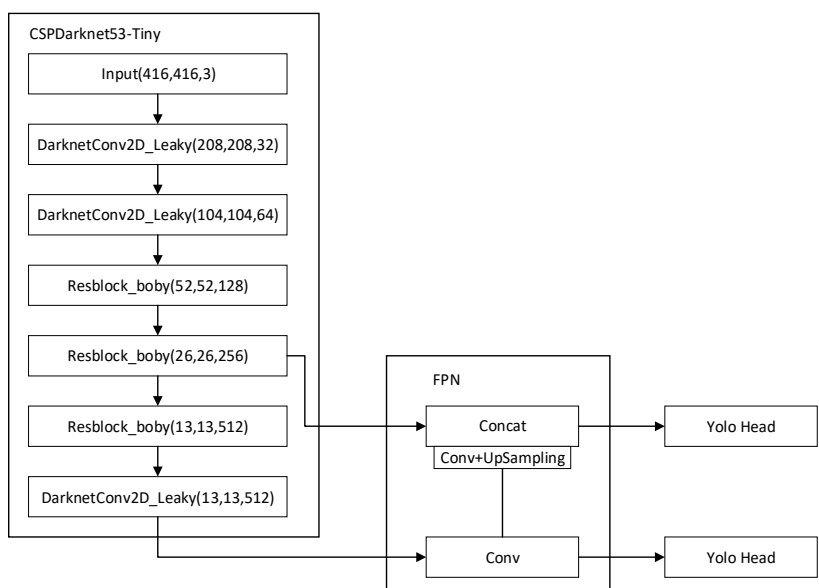

**Figure 2.** YOLOv4-tiny network structure.

### 3.1. Backbone Feature Extraction Network

Unlike CSPdarknet53 [31] networks, CSPdarknet53_tiny uses a new activation function, Leaky ReLU, which accelerates the iteration of the network at training time. Because of CSPdarknet53_tiny using the CSP net architecture, the complexity of the model can be greatly reduced, and the amount of computation can be saved by 20% [32]. In addition, channel segmentation is also performed on the image. The CSPdarknet53_tiny [33] network structure is roughly composed of three DarknetConv2d_BN_Leaky architectures and three Resblock_body architectures.

### 3.2. YOLOv4-Tiny Algorithm Flow

Firstly, the YOLOv4-tiny [34] algorithm process needs to collect and preprocess the data. Secondly, all images are labeled with labels for training and testing. Finally, the training results are applied after obtaining the training results. The algorithm flow chart is shown in Figure 3 below.

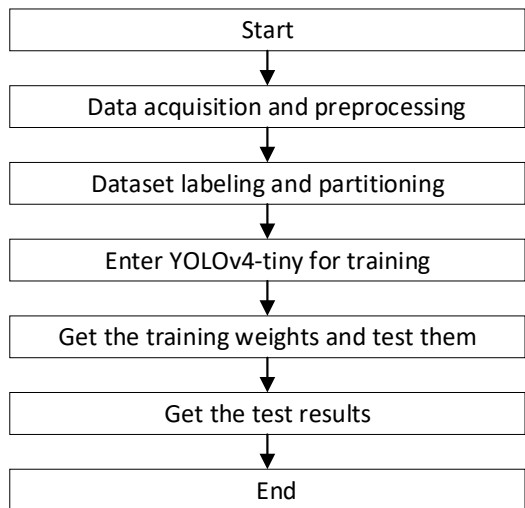

**Figure 3.** YOLOv4-tiny algorithm flowchart.

## 4. Detailed Design

### 4.1. Experimental Environment

The experimental environment of this study is the operating system Windows 1064-bit, developed by Microsoft (Redmond, WA, USA), used to run the experiment. The CPU is an Intel Core i7 9750HF, developed by Intel (Santa Clara, CA, USA). The GPU uses an NVIDIA Geforce GTX1650, developed by NVIDIA corporation (Santa Clara, CA, USA), graphics card. The deep learning framework is Pytorch, developed by Facebook AI Research (Menlo Park, CA, USA) (Table 1).

**Table 1.** Experimental environment software and hardware table.

| Name | Information |
|---|---|
| Memory | 16 GB |
| CPU | Inter(R)Core (TM)i7-9750HFCPU@2.60 GHz |
| GPU | NVIDIAGeforceGTX1650 |
| Operating system platform | Windows1064-bit |
| Deep learning computing framework | Pytorch1.13.1 |
| Development language | Python3.6 |

### 4.2. Make Datasets

#### 4.2.1. Dataset Format

In this study, the official VOC2007 dataset (the open dataset can be seen at http://host.robots.ox.ac.uk/pascal/VOC/voc2007/, accessed on 10 June 2023) was selected for pre-run. So, when making the dataset, the dataset was also organized in the VOC format to ensure its correctness. The VOC format is shown in Figure 4.

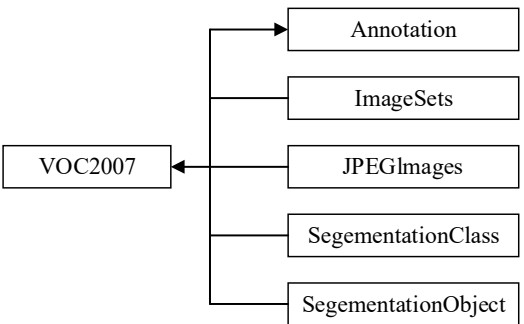

**Figure 4.** Standard VOC dataset format.

As shown in Figure 4, there are five directories in the root directory of the standard VOC format dataset. JPEGImages stores various types of images, which are used for training and testing, while annotations store tag files in an XML format related to JPEGImages. The ImageSets folder stores the txt files divided by the dataset. SegmentationClass stores the semantic segmentation result graph. SegmentationObject stores the image segmentation result map. The semantic segmentation result map and image segmentation result map are not used in object detection, so these two directories are not needed when making datasets in this study. The format of this research dataset is organized as shown in Figure 5.

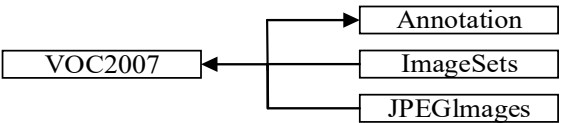

**Figure 5.** VOC dataset format.

### 4.2.2. Image Collection

Firstly, after web crawling and screening, about 600 apple images are selected as datasets for training and verification, as shown in Figure 6.

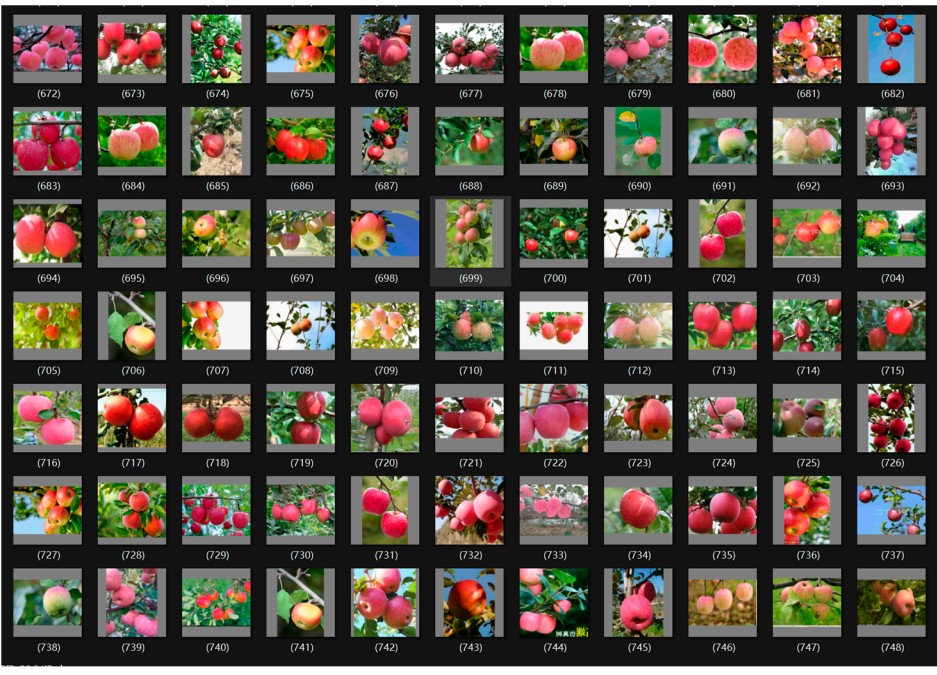

**Figure 6.** Filtered apple images.

### 4.2.3. Dataset Enrichment

In deep learning, to obtain better model performance, a large number of samples are usually required. However, if the number of samples is insufficient or the quality is not enough, the YOLOv4-tiny algorithm can improve the quality of samples by preprocessing the dataset, such as mosaic enhancement, rotation, scaling, translation, miscutting, etc., to make the model more generalizable. The following are several common ways to enrich datasets:

1. Image flip: instead of rotating the image 180° in a fixed direction, the image is flipped like a mirror, as shown in Figures 7–10.
2. Image rotation: the image is rotated clockwise or counterclockwise by 90 degrees or 180 degrees, as shown in Figures 11–14.
3. Image enlargement or reduction. Enlarging or reducing an image relative to the original image and changing its size can also augment the image.

4. Random cropping: randomly select a part from the image, crop this part, and adjust it to the size of the original image.

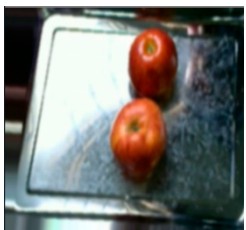

**Figure 7.** Original picture.

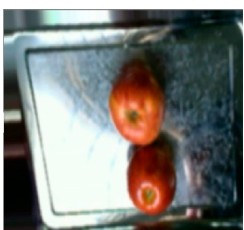

**Figure 8.** Original picture flipped vertically.

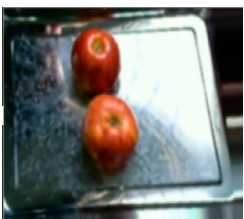

**Figure 9.** Original picture flipped horizontally.

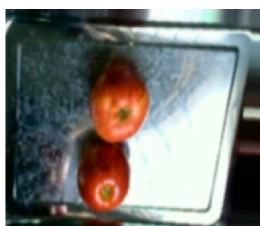

**Figure 10.** Original picture flipped vertically after horizontal flip.

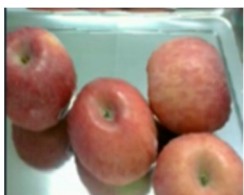

**Figure 11.** Original picture.

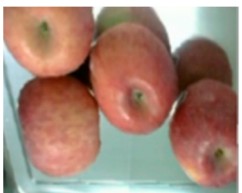

**Figure 12.** Original image rotated counterclockwise by 180°.

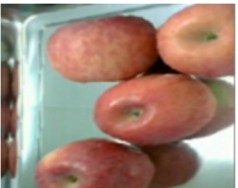

**Figure 13.** Original image rotated counterclockwise by 90°.

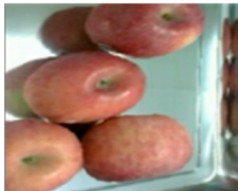

**Figure 14.** Original picture rotated 90° clockwise.

### 4.3. Image preprocessing

In this study, Labellmg (a powerful visual graphic identification application) is used for the manual annotation of the VOC label format, as shown in Figure 15. First, a rectangular box is drawn with the mouse on the collected picture to indicate the exact location of the fruit in that picture. The size of this rectangular box should be as close as possible to the size of the fruit, and if the occlusion rate of the fruit exceeds 70%, it should be avoided. In addition, if the fruit is at the edge of the image and the exposed area is less than 50%, it should also be avoided. When a series of fruits are labeled, Labellmg will automatically generate an XML label file. The <filename> of the image is recorded in the label in the XML file generated through manual annotation. <size> labels record overall information about the length, width, and number of channels of the image. Each <Object> label records the target information for each fruit in the picture, and its sublabel <name> records category information. The sublabel <bndbox> records the bounding box information and the target position information, including the coordinates of the upper-left and lower-right corners of the rectangular callout box. The file code is shown in the Supplementary Materials.

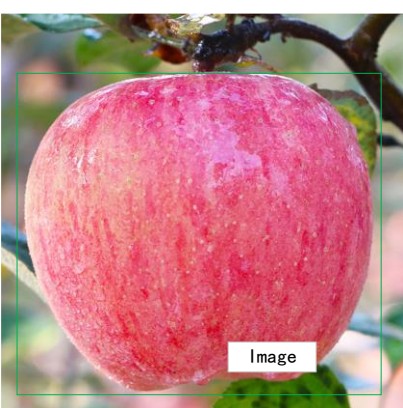

**Figure 15.** Labellmg annotation interface.

In this study, 2000 images are labeled as the final dataset. Among them, 1600 pictures are used as training sets, and 400 pictures are used as test sets. Considering the complexity of the real environment and stacked occlusion, 800 pictures of the 1600 positive samples in the training set and the images of apples without stacking occlusion are placed; 600 pictures with apple stacking occlusion are placed; and the remaining 200 pictures are negative samples without apples in the images. Similarly, the test set contains 350 positive samples and 50 negative samples. The positive and negative sample annotation plots are shown in Figures 16–21.

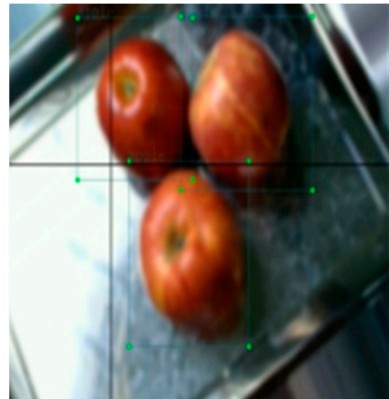

**Figure 16.** Example 1 of positive sample without stacking phenomenon.

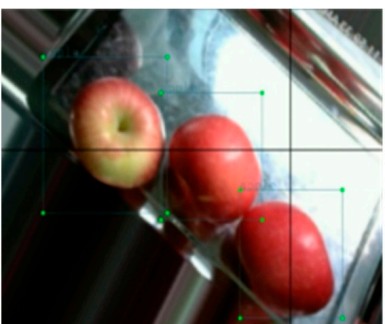

**Figure 17.** Example 2 of positive sample without stacking phenomenon.

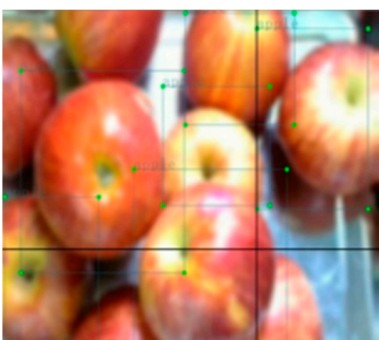

**Figure 18.** Example 1 of positive sample with stacking phenomenon.

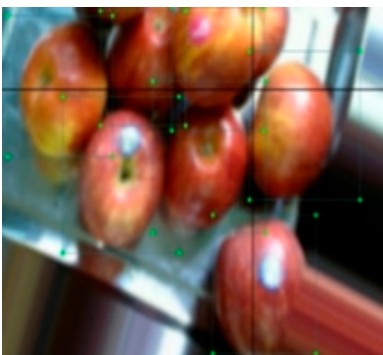

**Figure 19.** Example 2 of positive sample with stacking phenomenon.

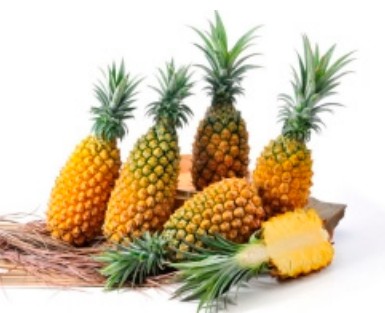

**Figure 20.** Example 3 of negative sample without test object.

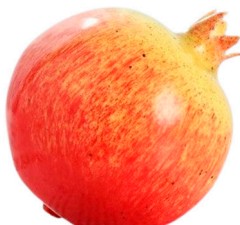

**Figure 21.** Example 4 of negative sample without test object.

In fact, negative samples are far more than the number of artificially placed samples in this study. In the training process, positive and negative samples are for the box generated by the algorithm after processing. Positive examples are used to make the prediction result closer to the real value, and negative examples are used to make the prediction result farther away from values other than the real value. Some boxes, even if they are labeled as positive samples, actually have a small IOU at training time, which will be treated as negative samples by YOLOv4-tiny. The most appropriate final ratios of positive and negative samples are 1:1 or 1:2. Neither involve far more negative samples than positive samples, which would result in the loss of positive samples, reducing the efficiency and detection accuracy of the network convergence, nor will they increase the false detection rate and false recognition rate because of too few negative samples.

## 5. Discussion of Results

### 5.1. Train the Model

In the preparation stage of YOLOv4-tiny model training, some hyperparameters are set in advance: the number of training rounds is 100, the batch size is 16, and the initial learning rate is 0.01. The final learning rate decays to 0.01 times the initial value, and the weight decay value is 0.0005.

### 5.2. Evaluation Indicators

The evaluation index is the objective basis for evaluating the quality of a model. The model can be quantitatively analyzed by calculating the evaluation index to carry out further parameter adjustment or ablation experiments. Before the evaluation indicators can be calculated, it is necessary to calculate the basic indicators in several binary or multiclassification tasks:

(1) TP (true positive): The positive samples are predicted as positive by the model. The evaluation criterion in YOlOv4-tiny is the number of detection boxes where the IoU of the prediction box and the real target frame is greater than or equal to a certain threshold, and the same real target frame is calculated only once;

(2) TN (true negative): the negative samples are predicted by the model as negative, that is, the number of background classes;

(3) FP (false positive): The negative samples are predicted by the model as positive, also known as false positive. It is evaluated in YOLOv5 as the number of detection boxes

whose IoU is less than a certain threshold between the prediction box and the real target box;

(4)  FN (false negative): a positive sample is predicted by the model as negative, also known as a false negative, that is, the number of real target boxes that are not detected.

Among the object detection technologies, precision, recall, F1Score, AP, and mAP are the five most widely used metrics.

Precision refers to the proportion of the target predicted by the correct model, which can be expressed by Equation (4).

$$P = \frac{TP}{TP + FP} \tag{4}$$

Recall, also known as the recall rate, refers to the proportion of all real targets that the model predicts correctly to the target, which can be expressed by Equation (5).

$$R = \frac{TP}{TP + FN} \tag{5}$$

F1Score is a harmonized mean of P and R, which can effectively balance precision and recall, calculated as shown in Equation (6).

$$F1\text{Score} = \frac{2P \cdot R}{P + R} \tag{6}$$

AP is a measure of average accuracy, which represents the area enclosed by the P-R curve. When R corresponds to multiple P values, only the maximum value is taken, and after PascalVOC2010, the P value corresponding to R will be regarded as the maximum value closest to its right curve. By setting $\bar{r}$ as the value of the independent variable of the curve on the right, the calculation formula is shown in Equation (7).

$$AP = \int_0^1 \max_{\bar{r} \geq r} p(\bar{r}) dr \tag{7}$$

mAP is the average category AP, which refers to the average value of each type of AP. It is mainly used to measure the advantages and disadvantages of multi-category target detection models and is one of the most commonly used evaluation indicators. Let the number of categories be c, and its calculation formula is shown in Equation (8).

$$mAP = \frac{1}{c} \sum_{i=1}^{c} AP_i \tag{8}$$

### 5.3. Analysis of Experimental Results

In this study, YOLOv4-tiny is used as a detector, trained on a homemade dataset of 2000 sheets, and obtained a series of evaluation metrics, including accuracy, recall, F1Score, and mAP. The specific analysis is as follows.

### 5.3.1. Loss Analysis

Taking the bounding box loss of YOLOv4-tiny training as an example, the loss image is shown in Figure 22.

As shown in Figure 22, the loss decreases sharply during the first 10 rounds of training. As the number of training rounds increases, it stabilizes by 90 rounds, indicating that YOLOv4-tiny essentially completed the fitting of the dataset. This can also be seen in other classification loss and confidence loss images, as shown in Figures 23 and 24.

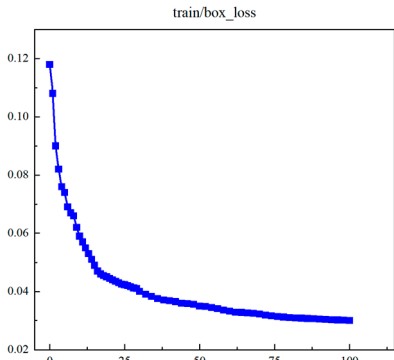

**Figure 22.** Bounding box loss for YOLOv4-tiny training.

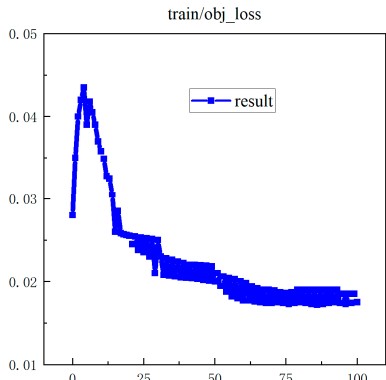

**Figure 23.** Loss of confidence image.

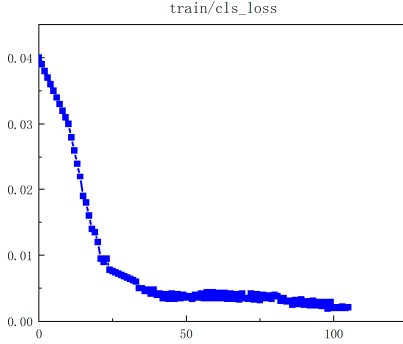

**Figure 24.** Classification loss images.

5.3.2. Precision and Recall Index Analysis

Accuracy analysis. The image accuracy is shown in Figure 25.

Precision refers to the proportion of the target predicted by the correct model. Therefore, the larger the value of the accuracy, the better, with one being the upper limit. It can be seen from Figure 25 that the accuracy of the model continues to climb when the number of training rounds increases. Finally, it will stabilize at about 80 rounds, fluctuating up and down between 0.9 and 1 (the best is 96.21%), indicating that the training effect is better.

With regard to recall analysis, the image recall is shown in Figure 26. The recall rate reflects the proportion of all true targets that the model accurately predicts the correct target. The higher the value, the better; one is ideal. As shown in Figure 26, as the number of training rounds increases, the recall climbs first, tends to level off at about 80 rounds, and finally is 95.47%. This also shows that the training results are better.

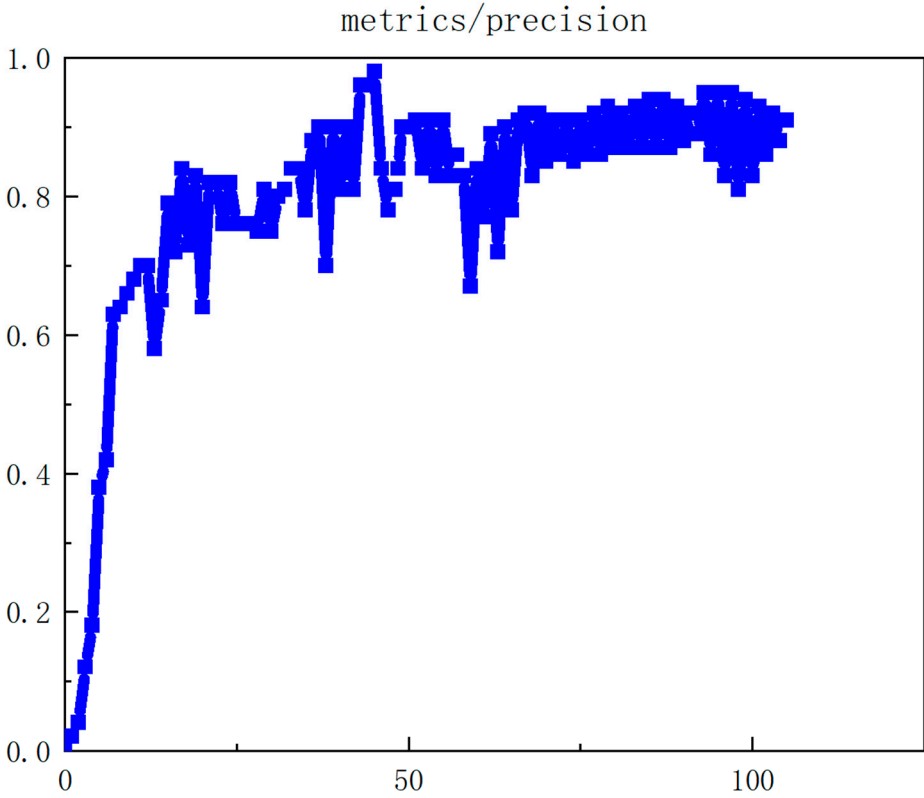

**Figure 25.** Image accuracy.

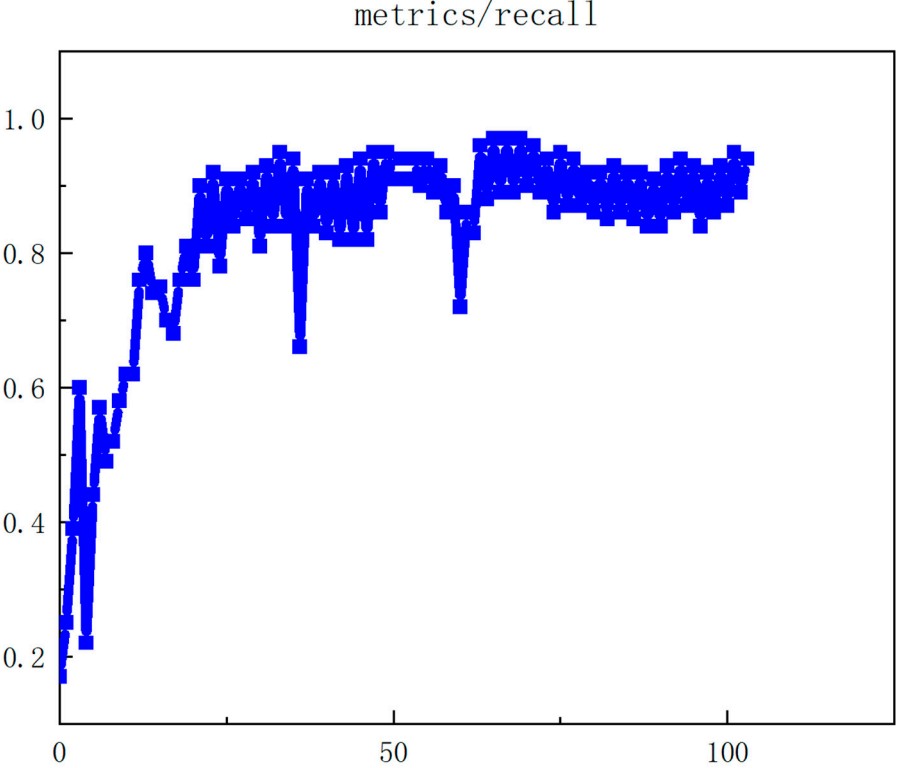

**Figure 26.** Image recall.

In terms of P-R curve analysis, the P-R curve is a two-dimensional curve, in which precision and recall are used as the coordinates of the vertical and horizontal axes, respec-

tively. The corresponding precision and recall at different thresholds are plotted as shown in Figure 27.

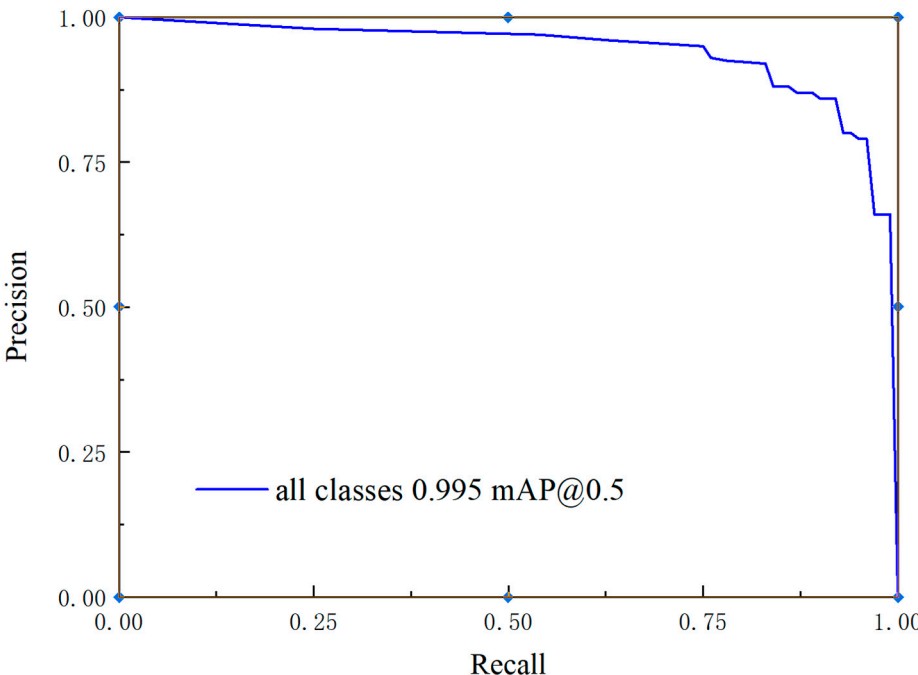

**Figure 27.** P-R plot.

In Figure 27, it is found that the recall decreases as the accuracy increases. When the accuracy drops to one, the probability score of the relevant positive sample is minimized. At the same time, the positive sample size divided by all sample sizes greater than or equal to the threshold is the lowest precision value. When the threshold is 0.5, the combined value of the P value and the R value reaches a maximum of 0.955, and the model training effect is the best.

### 5.3.3. F1Score Indicator Analysis

The image F1Score is shown in Figure 28.

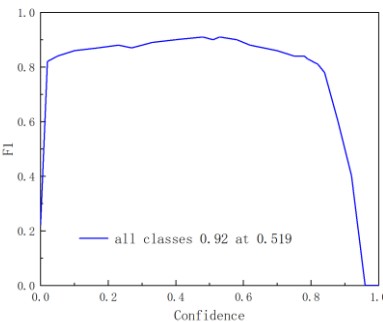

**Figure 28.** Image F1Score.

F1 represents the harmonic mean of P and R. As shown in Figure 28, the F1Score reaches a maximum of 0.92 when the confidence level is 0.519, indicating that the precision and recall of the model can reach the overall optimal value under this confidence level. In this case, the comprehensive performance of the model is the best.

### 5.3.4. mAP Analysis

The image of the mAP is shown in Figure 29.

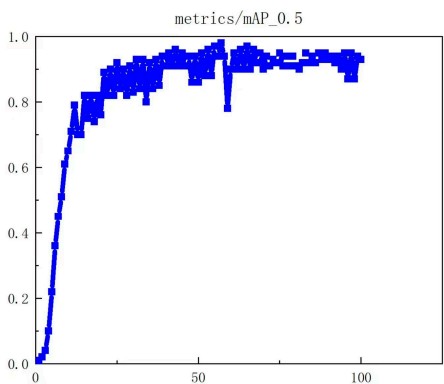

**Figure 29.** Image of mAP.

As shown in Figure 29, the mAP when the IoU threshold is 0.5 determines whether the detection frame matches the real target frame. It can be seen that the mAP indicator generally shows an upward trend and eventually converges (no less than 95.5%), indicating that the model has the best average detection effect for all categories.

5.3.5. Comparison Experiments

Compared with the traditional algorithm, the proposed method has some improvements, as discussed below:

(1) According Table 2, this study method's accuracy rate is up to 96%, while that of other methods is generally below 70%. The recall rate is 95%, showing a strong advantage compared with YOLOv4-Lite, YOLOv5-x, and Faster R-CNN. This situation is the same when applied to the F1Score and the mAP.

(2) The processing time (no more than 5 ms in this method) shows the greatest improvement when compared with the other methods (second level). Although the accuracy rate, the recall rate, the F1Score, and the mAP are related to many factors, such as the availability and effectiveness of the training picture, the complexity of the algorithm, etc., the compared algorithm factors will fluctuate with the factors mentioned above. So, the indicators (the accuracy rate, the recall rate, the F1Score, and the mAP) may not be the best representation of the efficiency of an algorithm, but the processing time is, because time is an absolute value. From the perspective of real-time performance, the method adopted in this study has incomparable advantages. The processing time is no more than 5 ms. Compared to 800+ ms, there is a huge improvement.

(3) Another result that needs to be analyzed is the size of the sample set; 2000 red fruit samples are used in this study, while more than 5000 are used in the comparison experiment. So, the algorithm in this study shows a certain overlearning phenomenon, and the too-high index value (more than 95%) is one of its manifestations. Therefore, the next step of this study is to continue to collect red fruit photos to form a larger sample set and avoid slight overlearning.

**Table 2.** Results comparison with Ref. [35].

|  | Accuracy Rate | Recall Rate | F1Score | mAP | Processing Time (s) |
|---|---|---|---|---|---|
| This study | 96.21% | 95.47% | 0.92 | 95.5% | 0.005 |
| Improved YOLO v4 [35] | 64.94% | 52% | 0.58 | 49.45% | 0.797 |
| YOLOv4 [35] | 64.21% | 43.51% | 0.52 | 44.1% | 0.848 |
| YOLOv4-Lite [35] | 35.33% | 27.39% | 0.31 | 18.57% | 0.88 |
| YOLOv5-l [35] | 51.5% | 32.75% | 0.4 | 27.06% | 0.879 |
| YOLOv5-x [35] | 53.33% | 34.65% | 0.42 | 30.17% | 0.871 |
| Faster R-CNN [35] | 51.61% | 16.48% | 0.25 | 10.81% | 0.955 |

**6. Conclusions**

To improve the efficiency and scope of agricultural operations, it is necessary to develop a general feature extraction model to overcome the limitations of the traditional image detection model that is currently limited by algorithms. This study focuses on apple picking and proposes a lightweight, coupon-product, neural-net terminal YOLO algorithm. The online real-time detection of this algorithm demonstrates good accuracy and high speed. It enables a more accurate capture of the appearance, structure, and characteristics of fruits. Using this algorithm for red fruit detection can aid in better apple classification, thereby improving accuracy, stability, and efficiency in grading, while reducing reliance on manual operations.

The shortcoming of this research is the dataset. The 2000-image dataset was built up by the research group, and is limited by the location and the seasons. Shortly, the team will share an open website with which to collect more apple images, which will enhance the algorithm and improve its accuracy. Furthermore, further investigation and potentially applying regularization techniques may be necessary to address this concern and improve the model's performance.

On the other hand, after the research of red fruit detection based on deep learning, it can be combined with binocular vision-related knowledge or tracking algorithms to achieve accurate fruit and vegetable positioning and picking. It can be widely used in modern agricultural production and has a wide development prospect.

**Supplementary Materials:** The following supporting information can be downloaded at: https://www.mdpi.com/article/10.3390/pr12010015/s1, Code S1: The core code of the real-time detection of red fruit based on the YOLO.

**Author Contributions:** Methodology, S.M.; Validation, W.D.; Formal analysis, J.W. All authors have read and agreed to the published version of the manuscript.

**Funding:** This work is supported by the demonstration and promotion project of equipment and technology in modern agricultural machinery, Jiangsu Province (NJ2021-18); Jiangsu Province Science and Technology Projects (BE2021016-2); the National Natural Science Foundation of China (32201681); the Ningxia Hui Autonomous Region Science and Technology Program (2021BEF02001); and The Fruit, Vegetable and Tea Harvesting Machinery Innovation Project of the Chinese Academy of Agricultural Sciences.

**Data Availability Statement:** Data are contained within the article.

**Conflicts of Interest:** The authors declare no conflict of interest.

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
