# Peer review of "Research on the Real-Time Detection of Red Fruit Based on the You Only Look Once Algorithm"

_processes, doi:10.3390/pr12010015_

Round 1
Reviewer 1 Report
Comments and Suggestions for Authors
This manuscript entitled "Research on Real-Time Detection of Red Fruit based on YOLO" developed a YOLOv4-tiny model for apple image recognition. Overall, in this study, the author collected image data and developed a YOLOv4-tiny algorithm for detection, but the algorithm was not improved by any methods and no real-time detection experiment was conducted. The paper should be much improved and the detailed comments are as follows.
1. In lines 55, 57, 93-96, and 131, the algorithms should be fully spelled at the first mentions.
2. In lines 142-143, does the author have any data or references to prove that YOLOv3 has become the most common deep learning object detection method today?
3. In line 159, the main research content of this study is missing.
4. Could the author explain why the input image was resized to 448*448*3 in line 198, but the input size was 416*416 in Fig. 3?
5. In line 221, the data 20% needs reference.
6. In line 312, what is "super number"? Does the author mean hyperparameter?
7. In line 322, could the author explain what is "Third item"?
8. In lines 352-407, the author described the prediction precision, recall, F1, and mAP. The author should provided more discussions about the detection results and improved performance compared to the traditional algorithm.
9. In lines 416-417, the author concluded that "The process time or the latency time is no more than 5ms, which is suitable and acceptable for the real time pick system". However, the result of 5ms was neither described nor shown in the results and discussion section.
10. The study focused on real-time detection, but the author didn't talk about any results related to real-time detection in the results and discussion section. The simulated or experimental results are highly recommended to be added to the manuscript.
Comments on the Quality of English Language
11. The author should pay attention to many typos in the manuscript. For example, capitalized "Classify" in line 213, lowercase "finally" in line 227, lowercase "the" in line 235, "Labelimh" in line 283, and so on.
12. The author should also correct many grammar errors throughout the manuscript, such as no subject in "draw a rectangular box" in line 278 and confused usage of tense in lines 362-363 and 370 (past tense, present tense, and future tense were totally misused), and so on.
Reviewer 2 Report
Comments and Suggestions for Authors
see the attachment

Moderate editing of English language required
Round 2
Reviewer 1 Report
Comments and Suggestions for Authors
The author didn't answer comments 8-10 in the point-to-point reply and I could not see any changes according to comments 8-10 in the revised manuscript.
Reviewer 2 Report
Comments and Suggestions for Authors
It seems author only responded in the cover letter but failed and forgot to reflect those changes in the manuscript.
It is must to incorporate all the changes in the manuscript rather than just responding to the comments in cover letter. Author also should be very careful about this negligence.
Comments on the Quality of English LanguageModerate editing of English language required
Round 3
Reviewer 1 Report
Comments and Suggestions for Authors
The authors provided new results based on various algorithms in Table 2 without any descriptions and discussions.
Author Response
Thank you for your kindly remind. The compare analysis about the result are added at the Table 2. Please check it at the manuscript.
"According the Table 2, the manuscript method accuracy rate is up to 96% while the other methods are generally below 70%. The recall rate is 95%, show strong advantage compared with the YOLOv4-Lite, YOLOv5-x and Faster R-CNN. This situation is the same applied on the F1Score and the mAP. What is more, the processing time (no more than 5ms in this method) show the greatest improvement when compared with the other methods(Second level). The details can be tracked in the Table 2."
Any information you want me know, please do no hesitate to email me.
Mei Song
Reviewer 2 Report
Comments and Suggestions for Authors
the author has incorporated all the comments asked in the previous review. the quality of the paper
Comments on the Quality of English LanguageMinor editing of English language required
Author Response
The English language has been modified. Please see the attachment.
